# Optimization of Nutrition after Brain Injury: Mechanistic and Therapeutic Considerations

**DOI:** 10.3390/biomedicines11092551

**Published:** 2023-09-17

**Authors:** Roy A. Poblete, Shelby Yaceczko, Raya Aliakbar, Pravesh Saini, Saman Hazany, Hannah Breit, Stan G. Louie, Patrick D. Lyden, Arthur Partikian

**Affiliations:** 1Department of Neurology, Keck School of Medicine, The University of Southern California, 1540 Alcazar Street, Suite 215, Los Angeles, CA 90033, USA; raya.aliakbar@med.usc.edu (R.A.); pravesh.saini@med.usc.edu (P.S.); hbreit@usc.edu (H.B.); 2UCLA Health, University of California, 100 Medical Plaza, Suite 345, Los Angeles, CA 90024, USA; syaceczko@mednet.ucla.edu; 3Department of Radiology, Keck School of Medicine, The University of Southern California, 1500 San Pablo Street, Los Angeles, CA 90033, USA; shazany@usc.edu; 4Department of Clinical Pharmacy, School of Pharmacy, The University of Southern California, 1985 Zonal Avenue, Los Angeles, CA 90089, USA; slouie@usc.edu; 5Department of Neurology, Department of Physiology and Neuroscience, Zilkha Neurogenetic Institute, Keck School of Medicine, The University of Southern California, 1540 Alcazar Street, Suite 215, Los Angeles, CA 90033, USA; plyden@usc.edu; 6Department of Neurology, Department of Pediatrics, Keck School of Medicine, The University of Southern California, 2010 Zonal Avenue, Building B, 3P61, Los Angeles, CA 90033, USA; apartiki@usc.edu

**Keywords:** nutrition, traumatic brain injury, stroke, status epilepticus, anoxic brain injury, neuroinflammation, malnutrition

## Abstract

Emerging science continues to establish the detrimental effects of malnutrition in acute neurological diseases such as traumatic brain injury, stroke, status epilepticus and anoxic brain injury. The primary pathological pathways responsible for secondary brain injury include neuroinflammation, catabolism, immune suppression and metabolic failure, and these are exacerbated by malnutrition. Given this, there is growing interest in novel nutritional interventions to promote neurological recovery after acute brain injury. In this review, we will describe how malnutrition impacts the biomolecular mechanisms of secondary brain injury in acute neurological disorders, and how nutritional status can be optimized in both pediatric and adult populations. We will further highlight emerging therapeutic approaches, including specialized diets that aim to resolve neuroinflammation, immunodeficiency and metabolic crisis, by providing pre-clinical and clinical evidence that their use promotes neurologic recovery. Using nutrition as a targeted treatment is appealing for several reasons that will be discussed. Given the high mortality and both short- and long-term morbidity associated with acute brain injuries, novel translational and clinical approaches are needed.

## 1. Introduction

Acute brain injury, including ischemic and hemorrhagic stroke, traumatic brain injury (TBI), anoxic brain injury and status epilepticus, are common causes of death and major disability both in the United States (US) and globally [1,2]. Although primary injury occurs at the time of disease onset and cannot be modified, secondary brain injury from biomolecular and clinical disease progression can be minimized to prevent neuronal death and preserve neurological function in survivors. In this context, early in-hospital medical interventions are important in modifying the pathological trajectory. Given the large burden of disease from acute brain injury [2,3], novel translational approaches are necessary to both understand the pathophysiology of disease and identify effective treatment targets that can be implemented in clinical practice.

In those with moderate-to-severe brain injury, nutritional optimization through the prevention of malnutrition and supplementation of select dietary therapies is an emerging field of interest to mitigate secondary brain injury and promote cellular recovery. Although proper nutrition is thought to promote recovery after brain injury, there are few evidence-based guidelines to standardize clinical practice. This is partially explained by a paucity of rigorous clinical trials in this field as well as conflicting results in the critical care literature [4]. Despite this gap, the prevention of malnutrition remains a clinical priority. Defined as a state of insufficient nutrition that leads to a change in body composition and diminished clinical function [4], malnutrition is a contributing factor in the development of short-term complications and poor functional recovery. In patients critically ill from brain injury, it may go underrecognized as an important modifiable risk factor to improve outcomes.

The purpose of this comprehensive review is to describe how malnutrition impacts the biomolecular mechanisms of secondary brain injury after TBI, stroke and other neurological disorders and how nutritional status can be optimized in this population. We will further highlight emerging therapeutic approaches, including specialized diets that aim to resolve neuroinflammation and metabolic crisis after acute brain injury by providing pre-clinical and clinical evidence that their use promotes neurological recovery. Using nutrition as a targeted treatment is appealing for several reasons that will be discussed. This review will provide a shared model of malnutrition and nutritional optimization after acute brain injury with the aim of expediting translational research towards implementation and the improvement of patient outcomes.

## 2. Recognizing Malnutrition and Clinical Importance after Acute Brain Injury

Malnutrition remains underrecognized in hospitalized patients, with approximately one-third developing malnutrition during their hospital course [5]. Its prevalence in the intensive care unit (ICU) is even greater, occurring in up to 78% of patients [6]. The high prevalence of malnutrition is partially explained by the catabolic state induced by cytokine-mediated inflammation of critical illness, whereby amino acids are mobilized from skeletal muscle fibers resulting in rapid muscle catabolism and a negative nitrogen balance [7]. Among patients with neurological injuries and TBI, malnutrition is particularly common in those with more severe injury as defined by a poorer Glasgow Coma Scale (GCS) score [8]. Physical immobilization, dysphagia and inadequate nutritional intake also contribute to the development of malnutrition in this population [6,7]. Vulnerable populations, including older adults, ethnic minorities, those with lower socioeconomic status and individuals hospitalized >7 days have higher odds of malnutrition, with an increased risk for developing malnutrition-related complications [5,6].

The recognition of malnutrition dates back five decades to the classic Butterworth’s publication *The Skeleton in the Hospital Closet*, which brought new attention to the prevalence and dangers of malnutrition among hospitalized patients [9]. Since this sentinel work, research has continued to expand recognition. A standardized set of diagnostic factors was released in 2012 by the Academy of Nutrition and Dietetics (AND) and the American Society for Parenteral and Enteral Nutrition (ASPEN) that relies on six characteristics: insufficient energy intake, unintentional weight loss, decreased muscle mass, decreased subcutaneous fat, fluid accumulation and decreased functional status [10]. The European Society of Clinical Nutrition (ESPEN) also uses an etiology-based classification system of malnutrition that incorporates markers of acute disease and inflammation [11]. More recently, the Global Leadership Initiative on Malnutrition (GLIM) was released in 2019; it used a two-step approach for malnutrition diagnosis including both phenotypic and etiological criteria [12]. The standardization of nutritional definitions is necessary to accurately measure the prevalence of malnutrition to better understand the scope of the problem and will aid in improving homogeneity in clinical research.

The early identification of high-risk patients and the recognition of developing malnutrition are vital to supporting metabolic needs after TBI, stroke or other acute brain injury. Among hospitalized patients, it is associated with adverse outcomes, including poor wound healing, higher infection rates, increased length of hospital stay, increased hospital readmission rates, increased mortality and greater cost utilization compared to those without malnutrition [13,14,15]. Critically ill patients account for the largest share of increased hospital costs among the malnourished, highlighting the necessity for both improved recognition and novel interventional approaches to prevent the negative consequences of malnutrition [16].

## 3. Pathophysiology of Malnutrition after Acute Brain Injury

The pathophysiology of acute brain injury is complex. Multiple mechanisms contribute to secondary brain injury, including ischemia, metabolic crisis, oxidative stress, immune dysfunction and blood–brain barrier (BBB) permeability. As we recently reviewed, neuroinflammation is the primary cellular mediator of secondary brain injury, both promoting and being sustained by these pathological mechanisms [17]. Post-injury, this leads to a mismatch between metabolic supply and demand that exacerbates neuronal injury and cellular death in vulnerable regions. While optimal nutritional therapy promotes cellular homeostasis and improved neurological recovery, providing nutrition below patient requirements exacerbates the mechanisms of secondary brain injury and results in short- and long-term complications (Figure 1). In this section, we will review the incidence of malnutrition in specific acute neurological diseases and will describe the relationship between it and other pathological pathways underlying secondary brain injury. Many different types of acute neurological injuries share similar pathophysiology, highlighting the potential of generalizable nutritional interventions to support neuronal recovery across a wide spectrum of disease.

### 3.1. Malnutrition in Specific Acute Neurologic Diseases

#### 3.1.1. Traumatic Brain Injury

Traumatic brain injury affects persons of all ages, genders and socioeconomic groups, accounting for 2.5 million US hospital visits annually [18], and is of global importance. Although nutritional optimization is universally accepted as being an important therapy, the full impact of malnutrition on TBI pathophysiology and recovery has not been characterized. In those with severe TBI, malnutrition occurs in up to 76% of patients admitted to the ICU and is associated with increased mortality and worse functional outcomes [19,20,21]. In combination, critical illness and malnutrition promotes a pro-inflammatory, catabolic and immunosuppressed state that results in systemic complications of central fever and hospital-acquired infection that are both independent predictors of poor outcome [22,23].

At a cellular level, biochemical changes begin within minutes of injury and progress over months-to-years [24]. The primary driver of secondary brain injury is the post-TBI inflammatory response that is mediated by activated microglia and the release of damage-associated molecular pattern (DAMP) proteins that facilitate the recruitment of immune cells to the site of injury to participate in early cellular reparative processes [17]. When neuroinflammation persists unmitigated, it promotes several pathological pathways that include BBB dysfunction [25], complement-mediated thrombogenesis and ischemia [26], mitochondrial dysfunction and oxidative stress [27,28]. Beyond the acute phase, microglia remain phenotypically activated [29], inducing long-term inflammation that serves as a link between acute illness and chronic neurodegeneration and symptomatic cognitive impairment and dementia [30]. TBI-induced immunosuppression is not fully understood but is hypothesized to be the result of the downregulation of leukocyte activity in response to DAMP release and inflammasome activation [31].

Current treatment options in TBI are limited. Given the association between malnutrition, inflammation and immunosuppression, nutritional optimization has been proposed as a promising therapy to restore post-TBI cellular homeostasis to prevent secondary brain injury [32]. There is emerging interest in the use of specialized diets and essential supplements that support the resolution of inflammation and promote immune system function in survivors of moderate-to-severe TBI [33]. Despite the extensive use of immune-enhancing nutrition in non-head-injured trauma patients, its role in TBI remains exploratory, with few prospective clinical trials conducted to date.

#### 3.1.2. Acute Ischemic Stroke

Ischemic stroke, the most common form of acute stroke, is caused by the interruption of blood supply to the brain primarily by arterial thrombosis, embolism or systemic hypoperfusion. The final common pathway of threshold oxygen and glucose deprivation is neuronal apoptosis [34]; however, early metabolic changes can be detected. Although not part of current standard practice, observational clinical studies have demonstrated that lactate levels measured from cerebrospinal fluid (CSF) can act as surrogate for anaerobic metabolism and metabolic crisis in the setting of oxygen and glucose deprivation, correlating well with stroke evolution and outcome [35]. As seen after TBI, DAMPs released after stroke activate local microglia and peripheral leukocytes and result in a massive release of pro-inflammatory cytokines [36]. Although this facilitates the clearance of cellular debris in infarcted areas, deleterious effects also occur.

Central nervous system (CNS) disease-induced immunosuppression in the setting of neuroinflammation is of special interest and has been extensively studied in stroke models. The hyperactivation of immune cells by the inflammasome results in the exhaustion of mature leukocytes and the release of immature leukocytes that contribute to the development of post-stroke immunosuppression [37,38]. In addition, catecholamine surges observed after ischemic injury induce T-cell lymphopenia and dysfunction that are detectable within hours of onset and precede the development of infection [39,40]. Additional stress responses characterized by autonomic and endocrine dysfunction further increase metabolic demand acutely after stroke.

The high prevalence and negative impact of post-stroke malnutrition, neuroinflammation and immunosuppression are of profound clinical importance. Ischemic stroke patients are at high risk of malnutrition during acute hospitalization due to cognitive impairment, a high occurrence of dysphagia and reduced physical mobility [41]. Protein-deficient malnutrition occurs in over 25% of hospitalized stroke patients within the first week of stroke and is associated with elevated cortisol levels, an increased rate of nosocomial infections and worse clinical outcomes [42]. Given these findings, it has been recently proposed that anti-oxidative and anti-inflammatory nutritional strategies are necessary to reverse or stop malnutrition and support neuronal plasticity to improve patient outcomes after stroke [43]; however, additional research is needed to better understand the importance of nutritional status in this population. Compared to TBI, acute ischemic stroke affects a relatively older population and is associated with greater motor and functional impairment [44,45]. Such factors may modify the response to dietary therapies and should be the subject of future clinical study.

#### 3.1.3. Acute Hemorrhagic Stroke

Spontaneous non-traumatic intracerebral hemorrhage (ICH) and subarachnoid hemorrhage (SAH) are hemorrhagic stroke types that are associated with worse clinical outcomes compared to ischemic stroke [46]. Although the biochemical pathology of hemorrhagic stroke has similar mechanisms to TBI and acute ischemic stroke, differences in metabolic demands likely exist. Resting energy expenditure (REE) and nutritional requirements are generally higher in critically ill ICH and SAH patients [47,48] and are thought to result from greater elevations in circulating catecholamines and increased inflammation-mediated protein catabolism [41]. In this population, nutritional status can predict post-ICH complications, including hematoma expansion and aspiration pneumonia, and is strongly associated with clinical outcomes [49]. It is unclear if hemorrhagic stroke patients benefit from a greater monitoring of energy expenditure and higher caloric or protein targets for nutritional support. Because of greater mortality and long-term morbidity associated with hemorrhagic stroke, this population is of special clinical interest.

#### 3.1.4. Anoxic Brain Injury

Mortality from cardiac arrest affects over one million people worldwide annually [50]. Anoxic brain injury is a common sequela of cardiac arrest, contributing to 68% of out of hospital deaths, and is associated with poor neurological recovery in survivors [51]. Nutrition plays a significant role in post-cardiac arrest recovery, as malnutrition is associated with increased mortality [52]. The optimal nutritional approach to improve outcomes in this population is unknown. In a population of post-cardiac arrest patients on ECMO circulatory support, Guitierrez et al. found that delayed enteral nutrition (>48 h from admission) was associated with improved neurologically favorable survival at discharge [53]. The authors hypothesized that premature feeding might result in vasoactive hormone release and a two-fold increase in splanchnic blood flow that could negatively impact brain perfusion.

Biomechanistically, the pathophysiology of anoxia involves a complex array of injury processes that include oxidative stress, mitochondrial dysfunction, and central hyperthermia [54]. Neuroinflammation is a primary mediator of secondary brain injury after cardiac arrest. Like other acute neurologic diseases, microglia become activated with secondary release of pro-inflammatory cytokines and free radicals, and recruitment of immune cells to sites of injury. Unlike stroke, CNS areas of injury may be diffuse or global. Mitochondrial dysfunction and metabolic crisis are also important disease features. In a mouse model of cardiac arrest, Ji et al. demonstrated that cerebral ischemia results in increases in the permeability of mitochondrial membranes and the production of reactive oxygen species (ROS) [55]. Vereczki et al. demonstrated that restoration of oxygen after transient deprivation attenuates oxidative stress and neuronal death in the hippocampus, highlighting the susceptibility of mitochondria and neurons to hypoxic injury [56].

Nutritional status is likely to impact these processes. In a mouse model of perinatal hypoxic-ischemic brain injury, Brandt et al. demonstrated that supplementation with various nutrient components including antioxidants was associated with reduced cytokine levels, microglial activation, the infiltration of immune cells and reduced lesion size [57]. In stroke and anoxic injury models, essential omega-3 (n-3) fatty acids downregulated the NF-kB pathway, resulting in the inhibition of pro-inflammatory cytokines and reduced damage by ROS [58]. Given the importance of essential lipid nutrients in maintaining cellular membrane function and integrity, their intake may further impact mitochondrial membrane composition and protect against Ca^2+^-induced apoptosis seen after cardiac arrest to mitigate mitochondrial failure and metabolic crisis [59].

#### 3.1.5. Status Epilepticus

Epilepsy and status epilepticus (SE) are common in hospitalized patients. With an annual incidence of up to 41 per 100,000 patients and an overall mortality of 20%, seizures and SE remain an important cause of acute brain injury [60]. Despite several advances in the study and treatment of SE, less is known regarding how nutrition impacts the biomechanisms of disease and treatment.

Chronic malnutrition has emerged as a significant factor in epilepsy development and seizure recurrence. This is especially true in developing countries, where malnutrition occurs in 22% of those with epilepsy [61]. Pre-clinical models suggest that nutritional deficiencies play a role in seizure development. In a rodent model, Palencia et al. demonstrated that the seizure threshold induced by pentylenetetrazol was lower in malnourished rats compared to nourished animals [62]. Further histological analysis in this study showed increased atrophic neurons in the hippocampus and cerebral cortex associated with malnourishment.

Acutely, SE increases superoxide species generation by nicotinamide adenine dinucleotide phosphate (NADPH) oxidase through N-methyl-D-aspartate (NMDA) receptor activation [63]. The overproduction of ROS and reactive nitrogen species during SE can lead to structural damage in mitochondrial membranes and resulting dysfunction. Several groups have therefore hypothesized that antioxidants are neuroprotective against seizure. One such agent, N-acetylcysteine (NAC), can replenish glutathione stores and promote antioxidant activity. In a pentylenetetrazol-induced seizure model, Devi et al. showed that NAC also exhibited an anticonvulsant effect [64]. Ubiquinone or Coenzyme Q plays a critical role in the mitochondrial electron transport chain and is an essential antioxidant and modulator of membrane permeability [65]. In pilocarpine-induced seizures, Tawfik et al. demonstrated that ubiquinone supplementation reduced oxidative stress and enhanced the antiepileptic effects of phenytoin [66].

In SE, BBB function is negatively affected by malnutrition and neuroinflammation. Dysfunction in the BBB is associated with astroglial injury, increased neuronal excitability and epileptogenesis [67]. Protein calorie malnutrition further exacerbates BBB dysfunction. In a rodent study by de Aquino et al., mice given low protein diets had increased ROS production, both systematically and in the CNS. Additionally, increased BBB permeability was evidenced by an increase in hippocampal brain tissue albumin content and astrogliosis [68].

Ketogenesis is an important compensatory mechanism in SE that is negatively impacted by malnutrition. Imbalances in ketosis have been hypothesized to contribute to disorders in brain energy metabolism and ion channel function. Through ketosis, the liver uses long-chain fatty acids to synthesize three ketone bodies: β-hydroxybutyrate (βHB), acetoacetate and acetone [69]. These substances serve as an alternative CNS energy source in the absence of glucose. It has also been suggested that ketogenesis can up-regulate the expression of energy metabolism genes and improve mitochondrial function [69]. Diets that promote ketone formation play a role in the treatment of SE, with the molecular basis of this therapy discussed below.

### 3.2. Considerations in the Pediatric Population

#### 3.2.1. Differences in Metabolism during Neurodevelopment

While the adult brain consumes 15–25% of total glucose metabolism during the resting state despite only weighing 2% of the human body, brain metabolic demands are far more dynamic during normal neurodevelopment and growth periods [70,71]. Combined positron emission tomography (PET) and magnetic resonance imaging (MRI) data demonstrate that brain glucose metabolism and body growth rate are inversely associated, implying that normal brain development requires a compensatory slowing of body growth rate across typical development [72].

Beyond adaptive glucose metabolism, other micro- and macronutrients sustain normal brain development and function. Lactoferrin is one such factor that plays a protective role in postnatal physiology. Released in colostrum, this protein is a major beneficial factor in breastfeeding. Lactoferrin’s ability to protect against neuronal injury and sepsis through anti-apoptotic, anti-oxidative and antimicrobial properties has been demonstrated in pre-clinical models of perinatal inflammation and neonatal hypoxia-ischemia [73]. Its propensity to augment neurotrophic compounds and brain connectivity might also improve cognition and neuropsychological function later in life after injury.

A major endogenous source of nutrients during neurodevelopment is the gut microbiota and its metabolic byproducts. Gastrointestinal tract microorganism colonization begins within a few hours of birth and typically matures by four years of age. Its makeup is determined in part by breastfeeding and other early nutritional factors and genetic conditions [74]. Of special interest, short chain fatty acids produced by microbiota help to modulate systemic immunological functions in addition to regulating inflammatory responses. Gut bacteria also produce several neuroendocrine hormones including gamma aminobutyric acid (GABA) and serotonin [75]. Given these critical byproducts, it is not surprising that gut microbiome composition and bacterial shifts induced by early growth, dietary changes including malnutrition and various disease states can influence the pathophysiology of CNS disorders.

#### 3.2.2. Genetic Factors

There is a paucity of studies evaluating the effect of genetic nutritional and metabolic factors on outcomes in children after brain injury. As in adults, the apolipoprotein E4 genotype has been associated with worse outcomes in children with moderate-to-severe TBI. In one study of 70 children, those with the ApoE4 genotype were more likely to exhibit symptoms and have unfavorable neurological outcome after TBI [76]; however, results have been inconsistent [77,78]. Genetic factors have also been investigated as therapeutic targets. One such genetic variable is the class of ATP-binding cassette (ABC) transporter protein genes which encode P-glycoproteins. These genes have been associated with clinical outcomes after TBI through various mechanisms, including eliminations of xenobiotics, impact on solute transport across the BBB, alterations in the CNS antioxidant reserve and other mechanisms involved in cerebral edema formation and mitigation [79]. Further study is needed to determine if novel pharmacotherapies can target genetic factors that increase susceptibility to secondary brain injury in the developing and pediatric brain.

#### 3.2.3. Estimating Nutritional Requirements in Pediatric Disease

Determining nutritional requirements early after pediatric brain injury is challenging, with discrepancies between measured and estimated energy expenditure (EEE) observed in children with TBI [80]. A prospective observational study of 13 mechanically ventilated children with severe TBI admitted to the pediatric ICU demonstrated that measured energy expenditure from indirect calorimetry was significantly lower at 70% of REE calculations [81]. This reduced value is likely explained by sedation, neuromuscular blockade, temperature control, antiseizure medications and early feeding practices. Overestimations of caloric needs could lead to overfeeding and adverse effects including increased carbon dioxide production deleteriously affecting respiratory and CNS systems. Conversely, inadequate nutritional support could lead to delayed wound healing, reduced respiratory drive, muscle weakness and depressed immune system function. To address these challenges, the standard use of indirect calorimetry measurements for REE in critically ill children should be considered.

## 4. Emerging Nutritional Interventions to Promote Early Brain Recovery

### 4.1. Determining Nutritional Needs to Support Metabolic Demands in Brain-Injured Patients

Determining a patient’s nutritional requirements is challenging, especially in the ICU, as both underfeeding and overfeeding can lead to poor clinical outcomes. The variable metabolic response seen in acute brain injury can further confound this process, with head-injured patients needing an estimated 140% or more of baseline metabolic expenditure due to accelerated protein catabolism and increased metabolic rate [82]. Although indirect calorimetry remains the gold standard method for the nutritional assessment of energy requirements in the critically ill population, logistical challenges in clinical practice exist [83]. Normal metabolic processes are further disrupted with the development of hypoproteinemia that impairs biochemical reparative processes and escalates secondary brain injury.

Appropriate nutrition prescription is essential in the treatment of acute brain injuries and takes into consideration the changing metabolic needs of individual patients. There has been increased interest in supplementing various nutrient components, whether in isolation or administered in combination, to modulate immune system function, resolve neuroinflammation and promote tissue repair after acute brain injuries. Table 1 highlights the main nutrients being investigated. The exact route, dose and combination of nutrients has yet to be determined given the lack of large-scale prospective clinical trials.

Patients with more severe injury from stroke, TBI or other diseases are often unable to take oral feeding, requiring the administration of enteral feeding formulations through a gastric tube. In clinical practice, specific diets and formulations are selected by a skilled dietary clinician based on clinical judgement as part of a multidisciplinary critical care team [84]. The main types of enteral nutrition (EN) include standard or polymeric formulas (whole, intact proteins), semi-elemental (peptide-based formulas), elemental formulas (amino-acid based, fully hydrolyzed) or specialized formulas (disease-specific). Elemental and semi-elemental EN is used in patients who are unable to rapidly digest polymeric proteins. Providing amino acids and proteins is especially important after traumatic injury to support tissue repair, wound healing and the biosynthesis of biologically active peptides [19,85]. No widely adopted guidelines or high quality prospective clinical trials have compared different EN formulations head-to-head in patients with acute brain injury, leading to variability in clinical practice. Emerging interests regarding specific EN approaches and interventions are discussed below.

After the initiation of EN, methods for increasing and monitoring nutritional adequacy have not been standardized. Clinically, achieving nutritional goals early to avoid malnutrition is paramount to supporting neurologic recovery. Taylor et al. reported a trend toward improved neurologic outcome 3 months after injury in patients who received EN at the goal rate early after admission, thus receiving a higher mean percentage of estimated energy requirements (good neurologic recovery in 61% versus 39%; *p* = 0.08) [86]. Recent observational studies indicate that patients often do not meet prescribed nutrient intake goals, as Chapple et al. noted considerable energy and protein deficits that persist after ICU discharge [87]. Protein requirements remain high in this population in the range of 1.5–2.5 gm/kg/day, depending on severity of injury and critical illness and other variables such as age, weight, temperature regulation and the use of paralytics and other coma-inducing agents [83,88]. Frequent reassessment is necessary to account for dynamic metabolic response changes that are expected.

Despite the evidence for the negative impact of malnutrition on critically ill patients with acute brain injury, the available data to support the precision monitoring of nutritional status in this patient population is lacking. Additional pre-clinical and clinical research is needed to define the optimal approach and monitoring strategies in patients with acute neurological disorders that will ensure proper nutritional support. Interventions that are widely available and feasible are most likely to be implemented as part of standard clinical practice.

### 4.2. Current Guidelines and Controversies

#### 4.2.1. Early Versus Late Feeding

Although definitions in the literature vary, early feeding is generally defined as the initiation of EN within 24 to 48 h of ICU admission with late EN referring to initiation >48 h. In clinical practice, early EN is often recommended in the ICU, though with limited supporting evidence [82,89,90]. The theoretical importance of early EN is to preserve gut mucosal integrity and function to reduce bacterial translocation and endotoxemia that contributes to systemic inflammation, prevent malnutrition and promote neurological recovery; however, due to inconsistency between studies, investigations have not yielded generalizable results. A recent Cochrane systematic review evaluated 7 randomized controlled trials (RCTs) comparing early versus late feeding in mixed ICU settings. Given study heterogeneity, the authors concluded uncertainty whether early EN resulted in a difference in 30-day mortality, infectious complications, feed intolerance or gastrointestinal complications [4]. Many of the studies that suggested benefit with early EN are observational trials or clinical trials with small sample sizes, while several larger prospective RCTs have suggested a higher mortality with early enteral feeding, perhaps due to a disruption of autophagy and resulting mitochondrial dysfunction [91]. The increased nosocomial infection risk observed in some studies may be related to gastrointestinal intolerance and occurrence of aspiration pneumonia.

In the severe TBI population, multiple studies have shown that early initiation of feeding is associated with reduced mortality and infection risk and improved neurologic recovery. The biomechanistic effect has not been confirmed, but early feeding is hypothesized to introduce essential exogenous substrates that promote the maintenance of visceral protein and fat and improve immunocompetence [8]. For general critically ill individuals who cannot take oral diet because of decreased level of consciousness or mechanical ventilation, early EN within 24–48 h or within 72 h is recommended by current ESPEN and ASPEN guidelines [83,89]. Similar recommendations are provided for neurologic disorders, but with limited evidence [92]. A study by the Brain Trauma Foundation (BTF) estimated that for early EN, every 10 kcal/kg/day increase in energy intake was associated with a 30–40% decrease in mortality risk [93]. Despite evidence that early EN is beneficial, current guidelines remain conservative, recommending feeding to basal caloric replacement by at least the fifth day and at most the seventh day post-TBI to reduce 2-week mortality [20,82]. Some controversies surrounding early EN in severe TBI include concern for aspiration risk and resulting pneumonia, the development of gut dysmotility resulting in feeding intolerance and increased metabolic demand.

#### 4.2.2. Parenteral Nutrition

Parenteral nutrition (PN) is designed to provide nutrients directly to blood through intravenous access in those with gastric dysmotility or absorptive disorders. Unless specifically indicated, EN is preferred to PN, as PN has been associated with increased infectious complications and longer hospital stays [89]. A large RCT in critically ill adults comparing the early initiation of PN <48 h from admission to late initiation after day 8 showed faster recovery, fewer infectious complications and less hospital resource utilization in the delayed PN group [94]. The initiation of PN should be considered within 5 to 7 days in patients who are severely malnourished or in whom EN is not feasible [89]. The cause of harm from ultra-early PN is not fully understood, but overfeeding with PN may increase the likelihood of bacterial infections [95]. PN has also been associated with hepatotoxicity and cholestasis, which has been hypothesized to be due to a disruption in the normal enterohepatic circulation induced by PN [96]. In clinical practice, patients prescribed PN require frequent monitoring of serum electrolytes and cardiac function, as rapid changes in intravascular volume and shifts in potassium, magnesium and calcium may contribute to critical illness cardiac dysfunction.

#### 4.2.3. Glucose Metabolism and Control

The brain plays a significant role in the homeostatic regulation of glucose metabolism [97], but this is frequently impaired after stroke, TBI or other neurologic injury. Metabolic stress and a hypercatabolic state are driven by a surge in glucocorticoids, catecholamines and glucagon, resulting in an increased REE by up to 200%. In the weeks after brain injury and other critical illness, hypermetabolism resulting in glycogenolysis and gluconeogenesis results in a state of hyperglycemia that exacerbates muscle protein catabolism and malnutrition [91,98]. Although glucose is vital to ATP production that energetically supports reparative biosynthetic pathways, excess hyperglycemia may be detrimental in several ways. At a cellular level, it may have direct effect on membrane lipid peroxidation [99] and has been observed in stroke models to induce gene expression that contributes to cortical spreading depression that exacerbates metabolic crisis in vulnerable tissue [100]. Insulin resistance may contribute to long-term hyperglycemia and the development of diabetes mellitus. Chronically, hyperglycemia may alter astrocyte metabolism and inhibit astrocyte proliferation [101].

Early clinical studies from general critically ill populations demonstrated that tightly controlling hyperglycemia with insulin was associated with improved outcomes [102]; however, the subsequent landmark randomized NICE-SUGAR trial showed no benefit of intensive glucose control versus moderate control in the general ICU population [103]. A post-hoc analysis of the neurotrauma population in the NICE-SUGAR trial demonstrated an increase in episodes of hypoglycemia in the intensive group, leading to recommendation for a more liberal strategy in this population. Similarly, no benefit was demonstrated with intensive glucose control after acute ischemic stroke in the SHINE RCT [104]. The cause of harm from intensive glucose monitoring is likely multifactorial. In severe TBI, tight glucose control can result in a detrimental metabolic response when measured in brain interstitial fluid [105]. This question, however, remains controversial, as conflicting results from a recent systematic review showed a borderline increase in worse neurological outcome in a conventional glucose group compared to the intensive group after TBI [106]. Current recommendations in the general critical care population are for a target glucose concentration of 6–8 mmol/L [89].

#### 4.2.4. Recommendations for Nutritional Support in Pediatric Patients

Although not universally standardized, some general recommendations regarding optimal nutritional support in the pediatric population exist. Current BTF Guidelines for the Management of Pediatric Severe TBI provide two primary recommendations: (1) early EN support within 72 h from injury is suggested to decrease mortality and improve outcomes, and (2) the standard use of immune-modulating diet is not recommended [107]. In a prospective, single-center trial by Briassoulis et al., the use of an immune-enhancing formula which included supplemental glutamine, arginine, antioxidants, and n-3 fatty acids versus a more standard enteral formula did not result in improved survival or rates of infection in pediatric severe TBI [108].

Like the adult population, the early use of PN is not supported. The Early versus Late Parenteral Nutrition in the Pediatric Intensive Care Unit (PEPaNIC) trial was a multicenter RCT which included 1440 term newborns to children 17 years of age admitted to a pediatric ICU, with about 8% diagnosed with TBI. The application of late PN resulted in fewer infections and a shorter duration of intensive care compared to early PN, regardless of age or diagnosis [109]. This finding was counterintuitive to the established practice of advising the immediate initiation of nutrition in neonates who were generally thought to have lower metabolic reserves with higher risk for hypoglycemia.

The deleterious role of hyperglycemia has also been demonstrated in pediatric populations. In TBI, elevated admission blood glucose levels are associated with both short-term complications and long-term outcomes [110,111,112]. After neonatal hypoxic-ischemic injury, hyperglycemia was more predictive of injury severity and the presence of structural imaging findings compared to hypoglycemia [113,114]. Specific blood glucose targets have not been established but should aim to limit hyperglycemia while avoiding hypoglycemic events that can also lead to secondary brain injury.

### 4.3. Emerging Nutritional Therapies to Support Neurologic Recovery

#### 4.3.1. Ketogenic Diet

While KD has been applied for decades in children with drug-resistant or rare metabolic causes of epilepsy to reduce seizures, emerging science supports its use after acute brain injury. KD consists of high amounts of fats, moderate amounts of protein and very low quantities of carbohydrates. The theoretical mechanistic benefits are several, including the provision of an alternative energy substrate in ketone bodies, beneficial gut microbiome shifts, membrane stability and repair by providing essential lipids, antioxidant effects and the prevention of mitochondrial dysfunction (Figure 2). In rodents, KD reduced seizures in kainic acid seizure models [115,116], protected against ischemic and hypoglycemic damage [117,118] and was neuroprotective in TBI and spinal cord injury [119,120]. The beneficial effect is partially explained by the ability of ketone bodies to resist oxidative stress and maintain mitochondrial stability [121,122]. In a rat controlled cortical impact model of TBI, KD given after injury increased ATP levels [119]. Together, these findings suggest that exogenous ketones could have an impact on maintaining metabolic homeostasis after injury.

Astrocytes play a pivotal role in the regulation of brain oxidative metabolism and may benefit from exogenous ketone delivery. Under normal physiology, owing to their perivascular location, astrocytes are a prevalent site of glucose uptake and metabolism. The lactate they produce participates in the astrocyte-neuron shuttle to support neuronal oxidative metabolism [123]. In addition to their high glycolytic capacity to yield lactate, astrocytes also produce significant amounts of ketone bodies. This raises the possibility that astrocytes can shuttle both lactate and ketone bodies as a substrate for neuronal oxidative metabolism to maintain brain energy homeostasis in the setting of excessive synaptic activity and ischemia that follows acute brain injury [124]. Increasing the body’s source of fatty acids and ketone bodies through KD might improve astrocyte-neuronal energy dynamics, leading to improved neurological recovery.

The anti-seizure effect of KD is mediated by increasing neuronal inhibition. This is hypothesized to occur through several mechanisms, including the alteration of GABA metabolism or GABA receptors. High levels of ketone bodies also decrease glutamate concentration in the synaptic cleft, resulting in reduced neuronal excitation [125,126]. In clinical study, evidence for utilizing KD in acute pediatric and adult refractory and super-refractory SE is mounting [127]. Because refractory SE, by definition, is pharmacoresistant to standard anti-seizure drugs, few treatment options may be available. A recent systemic review found that KD was successful in achieving the cessation of adult SE in 82% of patient cases from observational studies [128]. Clinical benefit may also be found when used in clinical syndromes of new onset refractory status epilepticus (NORSE) and related febrile infection-related epilepsy syndrome (FIRES) [129]. Parenteral KD given intravenously has been demonstrated in pediatric SE [130], but this has not been extensively studied in adults.

Research interest in the clinical efficacy of KD in other forms of acute brain injury is ongoing. Prospective clinical trials have been proposed for both acute spinal cord injury and TBI [131,132]. In a pilot feasibility study by Rippee et al., KD was feasible after mild TBI and alleviated post-concussive symptoms [133]. Currently, there are no rigorous studies of the clinical utility of KD after ischemic or hemorrhagic stroke and anoxic brain injury; however, given the proposed benefits, KD could find similar applications.

Many scientific questions and practical limitations should be considered. Patient characteristics, including genetic factors, age, gender and race may impact the response to KD. Regarding age, both preclinical rodent models and human observational studies demonstrate that age modifies the effect of KD on ketone body formation after TBI [134,135], which may partially explain its established efficacy in pediatric epilepsy. The optimal timing and duration of the intervention should also be defined. In an observational study of human TBI using invasive cerebral microdialysis (CMD) catheters, ketone body formation was greatest in the early fasting phase, with the authors suggesting that ultra-early feeding in this population may be cellularly detrimental to energetics [135]. The optimal level of ketogenesis and the best way to monitor response to therapy are unknown but should be elucidated. An additional observational clinical trial using CMD in TBI patients demonstrated that the degree of biochemical ketogenic shifts in the brain correlated with both short and long-term clinical outcomes [136]. Non-invasive modalities of monitoring CNS metabolic status may be useful in future research. In the absence of advanced monitoring techniques, KD should be initiated and titrated in coordination with highly skilled nutritional dietitians with experience and expertise in its application.

#### 4.3.2. Omega-3 Supplementation

The incorporation of n-3 fatty acid supplementation in addition to standard nutritional support is a novel approach to optimize ICU nutrition and promote recovery after acute brain injury. Formulations for EN that contain n-3 fatty acid have been developed for their anti-inflammatory and pro-immune properties. Their use in modern ICUs is increasing; however, their effect has not been rigorously studied in patients with stroke, TBI or other acute brain injury.

The biochemical properties of n-3 fatty acids provide rationale for its use. Endogenous bioactive lipids have diverse biological activities in regulating cell growth, adhesion, migration, signaling and cell death. Eicosanoid precursor fatty acids include n-3 polyunsaturated fatty acids (PUFAs) like eicosapentaenoic acid (EPA) and docosahexaenoic acid (DHA). Their metabolites, termed specialized pro-resolving lipid mediators of inflammation (SPMs), are important regulators of inflammation. The generation of pro-resolving SPMs from n-3 and n-6 PUFAs is complex but begins with the enzymatic activity of cytosolic and calcium-independent phospholipase A2 at the phospholipid bilayer. The formation of bioactive PUFAs can also be enhanced through dietary supplementation, with widespread sites of action (Figure 3). At the site of injury, SPMs promote the killing and clearance of pathogens, reduce leukocyte infiltration and stimulate the macrophage-mediated efferocytosis of cellular debris and apoptotic-sequestered neutrophils [137]. They additionally inhibit pro-inflammatory cytokine and chemokine expression while upregulating pro-resolving mediators. Although not fully characterized in acute brain injury, SPMs attenuate neuroinflammation and neuronal death in several models of neurological disease that include epilepsy [138], Alzheimer’s disease [139], SAH [140] and ischemic stroke [141]. The ability to resolve inflammation without compromising immune activation is one of the positive aspects of developing SPMs. Because eicosanoids are naturally occurring, the supplementation of PUFAs and other lipids is considered safe with a large therapeutic window [142].

Investigations on eicosanoid PUFAs in pre-clinical models are promising. In rat models of TBI, oral DHA administered pre-injury has been demonstrated to reduce oxidative stress, CD-68+ cells and caspase-3 levels and improve spatial learning [143,144,145]. Together, these results suggest a neuroprotective effect of DHA against inflammation-mediated oxidative stress and pro-apoptotic signaling with behavioral correlation after TBI. DHA has also been shown to reduce pro-inflammatory signaling, reduce infarct volumes, improve behavioral outcomes and maintain mitochondrial integrity in experimental stroke models [146,147,148]. Bioactive lipid supplementation may also promote synaptogenesis by promoting the generation of phosphatides, enhancing neurological plasticity [149]. In a rat model of neonatal hypoxic-ischemic injury, dietary n-3 PUFA protected against brain damage by activating the Akt pro-survival pathway [150].

The efficacy of n-3 and n-6 FAs or SPMs have not been rigorously tested in human populations. In TBI, the literature is limited to case studies or small series where high EPA and DHA supplementation was suggested as a potential contributor to good neurologic outcome [151,152]. To date, no large prospective clinical trials have investigated the use of n-3 or n-6 PUFAs to reduce secondary brain injury and improve clinical outcomes after acute TBI, stroke, SE or anoxic brain injury. Despite this limitation, the proposed mechanisms, pre-clinical data and apparent safety of n-3 FAs and other bioactive lipids warrant further investigation for therapeutic use in the neurocritical care population. Although the ideal n-3 or n-6 FA dosing is unknown, combined doses of EPA and DHA up to 5 g/day do not raise safety concerns in adults [153]. In US clinical practice, EN with n-3 PUFA supplementation is considered feasible, with several widely used, commercially available feeding formulations that contain enriched amounts of n-3 PUFAs. Fish oil containing n-3 PUFAs can also be supplemented individually. In the general medical and surgical ICU population, n-3 PUFA-enriched nutrition has been shown to reduce nosocomial infection risk and improve hospital outcomes [154,155], with research needed to replicate these findings after acute brain injury.

#### 4.3.3. Other Immunonutrition

In addition to n-3 fatty acid supplementation, the use of immunonutrition to modulate immune function has recently been investigated in specific diseases, including acute brain injuries [83]. Although no universal definition exists, immunonutrition typically consists of glutamine, arginine and ribonucleic acids used in combination with n-3 PUFAs. It is hypothesized that when supplemented together, the synergy between nutritional components provides greater clinical benefit than when provided individually, most notably demonstrated in oncology and surgical populations [156,157,158]. Mechanistically, the use of immunonutrition is proposed to attenuate the inflammatory response and deliver essential nutrients such as glutamine and arginine that are otherwise rapidly depleted during catabolism. Glutamine has multiple biological functions, including its role in cell proliferation, as an element of ATP biosynthesis and glycogenesis, the maintenance of acid-base balance and as an immunomodulator [156]. Arginine is a substrate for the synthesis of several cellular proteins, nitric oxide, polyamines, glutamic acid, ornithine, proline and creatine. Nucleotides and their metabolites additionally stimulate lymphocyte proliferation and alter the immune function of natural killer cells and macrophages to reduce infection risk [159].

Despite the proposed benefits of immunonutrition, it remains unclear which target diseases and populations are most likely to benefit from this intervention. Beyond cancer and trauma patients, the precise role of individual constituents of immunonutrition are under-investigated in stroke, TBI and other CNS injuries. In this context, the 2016 Society of Critical Care Medicine and the American Society for Enteral and Parenteral Nutrition (ASPEN) Guidelines for the Provision and Assessment of Nutrition Support Therapy in the Adult Critically Ill Patient suggest considering arginine-containing immune modulating formulations with EPA/DHA supplementation for patients admitted with TBI [83,155]. The major limitations of the available literature include small sample sizes, the use of varying immunonutrient combinations, inconsistency in the duration of use and an overall lack of study rigor. Future large-scale, multicenter, prospective RCTs that adopt standardized protocols are needed to strengthen the available literature.

#### 4.3.4. Other Dietary Components

Mineral elements, including but not limited to zinc and magnesium, have been investigated for their potential role in improving clinical outcomes in neurological diseases. While zinc deficiency may be harmful, supplementation has a controversial role in TBI given toxic levels that have been demonstrated following experimental injury and a possible contribution to excitotoxic cell death [160,161]. Magnesium is thought to have more of a protective role in preventing excitotoxicity and maintaining normal cellular function following neurological injury [161,162], but its supratherapeutic supplementation is not part of routine clinical care.

Research highlighting the importance of maintaining gut microbiota homeostasis in neurological injury supports the modulation of immune, inflammatory and metabolic processes through the microbiota–gut–brain axis [163]. The administration of probiotics and prebiotics can be prescribed to theoretically restore dysbiosis and as supplemental intervention in brain injury [163,164]; however, robust evidence from the literature is still lacking.

### 4.4. Advanced Diagnostics for Nutritional Assessment with Investigations in Brain Injury

#### 4.4.1. Novel Neuroimaging Modalities

Computerized tomography (CT) remains the initial diagnostic imaging modality of choice after acute brain injury due to its rapid acquisition; however, novel MRI modalities can provide a comprehensive assessment of brain anatomy, structural changes, anatomical and functional connectivity and metabolic characteristics. Advanced techniques that have been studied in acute brain injury include enhanced structural imaging, diffusion tensor imaging, vessel wall imaging, functional MRI, MR spectroscopy (MRS) and chemical exchange saturation transfer (CEST) imaging. These novel modalities remain largely investigational but can provide insight into the metabolic changes underlying secondary brain injury after stroke, TBI or other acute neurological disorders. Advanced neuroimaging techniques can also serve as real-time in vivo biomarkers in future translation research.

MRS evaluates cerebral neurochemical status based on chemical shift and molecular environment and can detect both early and chronic changes. In a study of 32 patients with hyperacute MRS performed after acute ischemic stroke, elevations in lactate were measured even prior to the appearance of diffusion-weighted lesions that are the current gold standard for stroke diagnosis [165]. After acute severe TBI, membrane marker choline (Cho) increases and neuronal marker N-acetylaspartate (NAA), thought to be associated with mitochondrial function, decreases; notably, the improvement or persistence of a decreased NAA/Cho ratio in the chronic setting correlates with functional neurological recovery [166,167]. In future research, MRS might also be used to determine response to nutritional therapy. Glutathione (GSH), abundant in the brain, is crucial in maintaining redox homeostasis in the CNS through antioxidant activity. Studies of epilepsy patients on KD have demonstrated higher levels of brain GSH in response to the diet [168,169]. Other ketone bodies including acetone, βHB and acetoacetate can also be measured by MRS in future applications.

PET imaging can also assess nutritional and metabolic status; however, it is not widely used for this purpose. Several tracers have been studied to detect specific CNS targets, most notably oxygen and glucose metabolism, cellular amino acid uptake and the activation of microglia and inflammation [170]. After cardiac arrest, brain glucose metabolism shows the greatest reduction in cortical areas, suggesting their higher susceptibility to anoxic injury and metabolic crisis after brain injury [171,172]. Brain inflammation is measured by translocator protein (TSPO), which is involved in neurological inflammatory reactions. In rodent models, TSPO tracing has been used to demonstrate inflammation after ischemic stroke and response to therapy [173,174]. Like MRS, PET has been used to characterize the cellular effect of KD. A study by Horowitz et al. showed globally reduced brain glucose metabolism with KD, confirming the utilization of alternative energy sources with the intervention [175].

There is a paucity of data to support the use of other novel neuroimaging modalities for nutritional and metabolic assessment. A small study of 17 patients with mixed acute and chronic TBI demonstrated that CEST imaging could be used to detect cerebral acidosis secondary to excitotoxicity or ischemia, with results directly correlated with functional outcome [176]. In pre-clinical research, resting functional MRI has demonstrated that nutritional intake can alter functional network organization [177]. Despite the potential of novel neuroimaging techniques to monitor nutritional status and metabolic states, these technologies are not widely available and are limited by feasibility concerns in the ICU; however, they will remain an important tool in understanding how nutritional status can improve energetics and neuroplasticity after brain injury.

#### 4.4.2. Cerebral Microdialysis

Cerebral microdialysis is an invasive device placed in brain parenchymal tissue to continuously sample extracellular interstitial fluid to measure specific important metabolites, including glucose, lactate, pyruvate and glutamate. It is a powerful research and clinical tool to assess the metabolic state of the CNS that previously could not be directly monitored in vivo in continuous fashion. The current role of CMD is as a clinical research tool and as part of cerebral multimodality monitoring following severe brain injury. The goal of CMD and multimodality monitoring is the early detection of supply and demand mismatch and resulting metabolic crisis to allow for prompt reversal and prevention of secondary brain injury [178]. As previously discussed, following acute injury, the brain shifts from metabolizing glucose, its preferred fuel source for oxidative phosphorylation, to alternative substrates that include ketone bodies, lactate, glycerol and amino acids [91]. Several clinical studies have shown that an increased lactate:pyruvate ratio and hypoglycorrhachia precede clinical complications such as ischemia, seizures and intracranial hypertension [179,180]. Signs of metabolic crisis, including an elevated lactate:pyruvate ratio and low extracellular glucose, especially in the setting of concurrent hypoxemia, is strongly associated with worse functional outcomes and mortality in brain-injured patients [179,181,182,183].

Evidence of the influence of nutritional therapy on CMD parameters is inconsistent and lacking in the neurocritically ill patient population [180]. In a retrospective study of SAH patients with CMD placed, Schmidt et al. found no clear association between nutritional therapy and CMD glucose parameters; however, intensive insulin treatment for hyperglycemia was observed to reduce CMD glucose regardless of baseline serum glucose values [184]. Conflicting results have been reported in other SAH populations. Kofler et al. demonstrated that EN increased both serum and CMD glucose levels [185]. Regardless of differing results, these studies suggest the feasibility of using CMD to monitor the biochemical response to nutritional therapies. To date, similar studies have not been performed in acute TBI or stroke populations. Additionally, no studies have addressed the direct question of whether CMD-guided nutritional interventions impact clinical outcomes after acute brain injury.

## 5. Conclusions & Future Aims

Emerging science continues to establish the detrimental effects of malnutrition and consequent neuroinflammation on the post-injury brain. Given this, novel clinical approaches targeting nutrition are needed. Proposed interventions, including KD, n-3 PUFA supplementation and immunonutrition, have extensive pre-clinical data demonstrating efficacy in alleviating inflammation, promoting metabolic homeostasis, reducing oxidate stress and attenuating pro-apoptotic signaling; however, rigorous study in prospective clinical trials is needed to determine if proposed interventions will have clinical effect. Nutritional formulations to support anti-inflammatory and immune system function are widely available, relatively inexpensive and largely considered safe, providing further rationale for their study and implementation into standard clinical practice. This is especially important in resource-limited environments, including global and military populations, where novel and feasible treatment options are desired.

Many questions remain. Of greatest scientific priority, patient factors, including age, sex and other genetic and environmental factors that both predispose individuals to malnutrition as well as predict response to therapy, should be delineated. The further identification of novel nutritional biomarkers that accurately measure nutritional status or predict clinical outcomes will also aid in future translational research. Clinically, greater clarity is needed regarding which patient populations will most benefit from select nutritional interventions, as well as the timing, dose and duration needed to significantly impact clinical outcomes. Given the short and long-term morbidity and mortality associated with acute brain injury, novel and pragmatic clinical study is urgently needed to identify and implement optimal nutritional strategies that will improve CNS function.

## Figures and Tables

**Figure 1 biomedicines-11-02551-f001:**
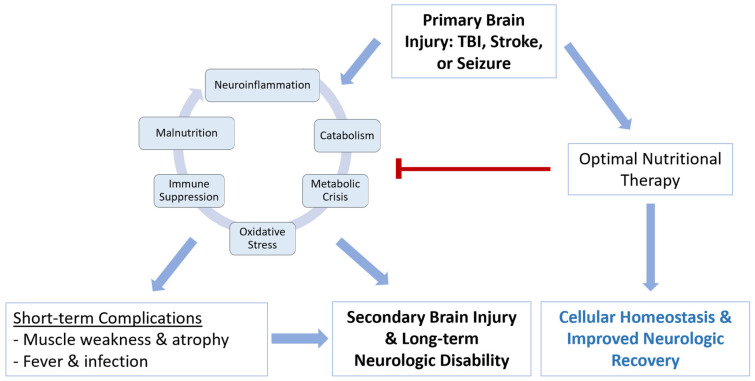
Conceptual schematic of the influence of nutritional status on mechanisms of secondary brain injury and recovery after acute brain injury.

**Figure 2 biomedicines-11-02551-f002:**
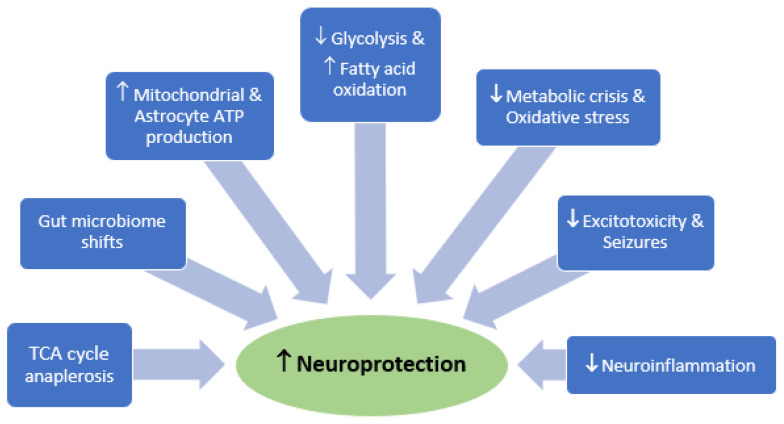
Interconnected mechanisms through which ketogenic diet confers broad neuroprotective effects in models of brain injury.

**Figure 3 biomedicines-11-02551-f003:**
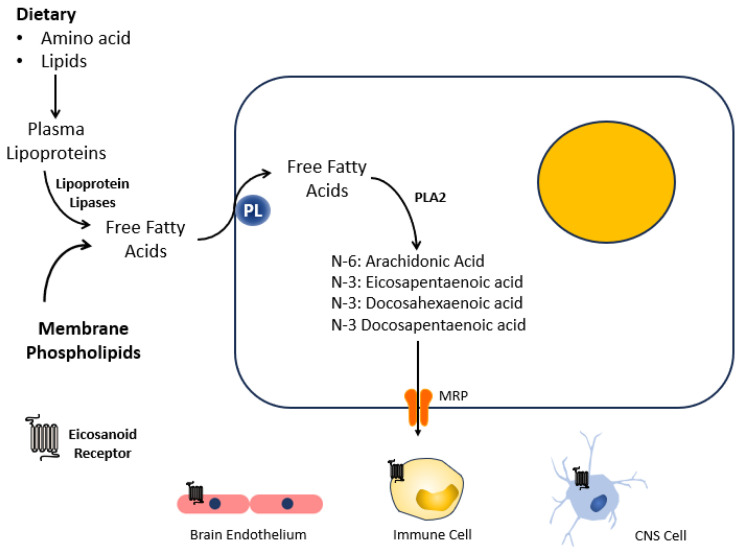
Generation and Sites of Activity of Eicosanoid Lipids. PL: phospholipid transporter, PLA2: phospholipase A2, MRP: multidrug resistance protein transporter, CNS: central nervous system.

**Table 1 biomedicines-11-02551-t001:** Specialized nutritional components and diets with a hypothesized role in promoting cellular homeostasis and neurological recovery after acute brain injury.

**Nutrient**	**Proposed Metabolic or Cellular Benefit**
Ketone bodies and/or ketogenic diet	Provision of an alternative energy substrate through ketone bodies. Reduced excitotoxicity, reduced oxidative stress, improved mitochondrial and astrocyte ATP production, anti-inflammatory and anti-seizure properties.
Omega-3 fatty acids: EPA, DHA, DPA	Promotion of the resolution of inflammation. Attenuated excitotoxicity, cell membrane stabilization, support of neurogenesis and antioxidant activities.
Glutamine	Promotion of cell proliferation, ATP biosynthesis, glycogenesis, maintenance of acid-base balance, immunomodulator.
Arginine	Synthesis of intracellular proteins, nitric oxide, polyamines, glutamic acid, ornithine, proline and creatine, immunomodulator.
Nucleotides	Stimulation of lymphocyte proliferation, modulation of the immune activity of natural killer cells and macrophages.
Other amino acids	Support for glutathione biosynthesis for tissue repair, support for catecholamine biosynthesis, ROS scavenging.
Mineral elements (ex. zinc, magnesium)	May contribute to regulating antioxidant capabilities.
Probiotics and Prebiotics	Support for the production of neuroactive molecules that promote communication between the gut microbiota and brain while maintaining homeostasis.

EPA: eicosapentaenoic acid; DHA: docosahexaenoic acid; DPA: docosapentaenoic acid.

## Data Availability

Not applicable.

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
