# Peer review of "Optimization of Nutrition after Brain Injury: Mechanistic and Therapeutic Considerations"

_biomedicines, 2023, doi:10.3390/biomedicines11092551_

Round 1

Reviewer 1 Report

The present manuscript reviews the recent progress about mechanistic and therapeutic considerations involved in optimization of nutrition after brain injury. Generally speaking, the manuscript is well prepared for and written. However, as a review, I recommend that authors can draw several figures (such as mechanistic figures) to increase the readability, or make their meaning more intuitive. I also have several comments, as followed below:

Abstract: good

Introduction: What is nutritional optimization? and what is malnutrition? Authors should give an exact definition in Introduciton section since they are the focus of the present review.

2. Line 65: for the first appearance, TBI should be provided its full name.

3. Figure 1, the figure is not clear with the black fond and blue background. Authors can delete the blue background or changed to other colors.

4. For the section of malnutrition in specific acute neurological diseases, authors give many words to describe their features and occurrence of several nuerological diseases. In my opinion, authors can give some figures to summarize them, which increase the readability of the manuscript.

5. In table 1, authors summarized the specialized nutrients in promoting cellular homeostasis., which mainly included amino acid and fatty acids. As a matter of fact, mineral elements are also important nutrients to affect neurological disease and there are many researches invovled in these aspects. Authors can include mineral elements in Table 1.

6. For the section of Emerging nutritional therapies to support neurologic recovery, authors included ketogenic diet,  amino acid and fatty acids. Similarly, mineral elements are also emerging nutritional terapies to support neurologic recovery, and there are many studies involved in these aspects. Authors can include mineral nutrition in the section. I also dislike the word immunonutrition since no universal definition for immunonutrition exists.

7. References: Some references are very old and seem un-relevant with the manuscript. Authors can delete them or replace them with new ones, such as references 9, 

In addition, the format for the references are not unanimous, such as reference 50, 55. There are abbreviations and full names for the journal names. The site for publication year varied. Authors should carefully check them and correct these mistakes.

English language seems generally good.

Author Response

The present manuscript reviews the recent progress about mechanistic and therapeutic considerations involved in optimization of nutrition after brain injury. Generally speaking, the manuscript is well prepared for and written. However, as a review, I recommend that authors can draw several figures (such as mechanistic figures) to increase the readability, or make their meaning more intuitive. I also have several comments, as followed below:

Figure #2 and Figure #3 were added to the “Nutritional Therapies” sections to more intuitively describe aspects of select specialized diets.

Abstract: good

Introduction: What is nutritional optimization? and what is malnutrition? Authors should give an exact definition in Introduciton section since they are the focus of the present review.

What is meant by “nutritional optimization” is now explained in line 54 and 55.

“Malnutrition” was/is defined in line 61-63.  A small edit was made for clarification.

  1. Line 65: for the first appearance, TBI should be provided its full name.

“TBI” was already defined in line 45.

  1. Figure 1, the figure is not clear with the black fond and blue background. Authors can delete the blue background or changed to other colors.

The color of the text boxes were changed to improve readability.

  1. For the section of malnutrition in specific acute neurological diseases, authors give many words to describe their features and occurrence of several nuerological diseases. In my opinion, authors can give some figures to summarize them, which increase the readability of the manuscript.

Figure 1 was slightly revised to communicate that primary brain injury encompasses more than just TBI.  We wanted to keep the focus on nutrition rather than detail the pathophysiology of different acute neurologic diseases, which is beyond the scope of this review.

  1. In table 1, authors summarized the specialized nutrients in promoting cellular homeostasis., which mainly included amino acid and fatty acids. As a matter of fact, mineral elements are also important nutrients to affect neurological disease and there are many researches invovled in these aspects. Authors can include mineral elements in Table 1.

Mineral elements was added to Table 1 and a brief paragraph was written under the “Nutritional Therapies” section.

  1. For the section of Emerging nutritional therapies to support neurologic recovery, authors included ketogenic diet, amino acid and fatty acids. Similarly, mineral elements are also emerging nutritional terapies to support neurologic recovery, and there are many studies involved in these aspects. Authors can include mineral nutrition in the section. I also dislike the word immunonutrition since no universal definition for immunonutrition exists.

We agree that there are limitations to the word/definition of “immunonutrition”, which we acknowledge in the “Other Immunonutrition” section (Line 639-640); however, in clinical practice, the use of the term “immunonutrition” is common, which is why it is used in this manuscript.

Mineral elements was added to Table 1 and a brief paragraph was written under the “Nutritional Therapies” section.

  1. References: Some references are very old and seem un-relevant with the manuscript. Authors can delete them or replace them with new ones, such as references 9.

Reference 9 is old, but it is historically important and adds to the narrative; therefore, it was included.

In addition, the format for the references are not unanimous, such as reference 50, 55. There are abbreviations and full names for the journal names. The site for publication year varied. Authors should carefully check them and correct these mistakes.

We have reviewed the references and used a consistent citation style.

Reviewer 2 Report

In this manuscript, the authors have provided a comprehensive overview of nutritional aspects relevant to neurological recovery after acute brain injury. The manuscript is well-written and engaging to read. However, some minor corrections should make before considering its acceptance for publication.

Based on the manuscript content, the Abstract should be mentioned the pediatric population.

Page 2, lines 53-55: add reference(s)

Page 5, lines 195-196: add a reference

In the title of Table 1, consider replacing “specialized nutrients” with “nutritional interventions” or removing ketone bodies/ketogenic diet from the Table.

Page 9; lines 370-382: there is no reference through the whole paragraph

Page 11, line 485: please rephrase the subtitle to reflect provided content

Page 11, lines 514-515: this sentence should be rephrased

Page 13, lines 580-582: this sentence is understandable and should be rephrased. Consider using the term eicosanoid precursor fatty acids. Additionally, throughout the whole manuscript, uniform fatty acid nomenclature (omega or n)

Page 13, lines 594-595: add reference(s)

Are there data on the doses for PUFAs used in clinical studies?

Lines 621-649: replace “immunonutrition” with “immunonutrients.”

Author Response

In this manuscript, the authors have provided a comprehensive overview of nutritional aspects relevant to neurological recovery after acute brain injury. The manuscript is well-written and engaging to read. However, some minor corrections should make before considering its acceptance for publication.

Based on the manuscript content, the Abstract should be mentioned the pediatric population.

A small edit was made to mention pediatric populations.

Page 2, lines 53-55: add reference(s)

This is a general statement that we believe is supported during the remainder of the manuscript.

Page 5, lines 195-196: add a reference

The following references were added to the sentence “Compared to TBI, acute ischemic stroke affects a relatively older population and is associated with greater motor and functional impairment.”:

Eng JJ, Rowe SJ, McLaren LM. Mobility status during inpatient rehabilitation: a comparison of patients with stroke and traumatic brain injury. Arch Phys Med Rehabil. 2002 Apr;83(4):483-90. doi: 10.1053/apmr.2002.31203. PMID: 11932849; PMCID: PMC3478323.

McCrea MA, Giacino JT, Barber J, et al. Functional Outcomes Over the First Year After Moderate to Severe Traumatic Brain Injury in the Prospective, Longitudinal TRACK-TBI Study. JAMA Neurol. 2021;78(8):982–992. doi:10.1001/jamaneurol.2021.2043

In the title of Table 1, consider replacing “specialized nutrients” with “nutritional interventions” or removing ketone bodies/ketogenic diet from the Table.

The Table 1 title was edited to be more inclusive.

Page 9; lines 370-382: there is no reference through the whole paragraph

The following citations have been added to this paragraph:

Patel JJ, Mundi MS, Taylor B, McClave SA, Mechanick JI. Casting Light on the Necessary, Expansive, and Evolving Role of the Critical Care Dietitian: An Essential Member of the Critical Care Team. Crit Care Med. 2022 Sep 1;50(9):1289-1295. doi: 10.1097/CCM.0000000000005607. Epub 2022 Aug 15. PMID: 35984051.

Dijkink S, Meier K, Krijnen P, Yeh DD, Velmahos GC, Schipper IB. Malnutrition and its effects in severely injured trauma patients. Eur J Trauma Emerg Surg 2020;46(5):993-1004. DOI: 10.1007/s00068-020-01304-5

Arribas-López E, Zand N, Ojo O, Snowden MJ, Kochhar T. The Effect of Amino Acids on Wound Healing: A Systematic Review and Meta-Analysis on Arginine and Glutamine. Nutrients. 2021 Jul 22;13(8):2498. doi: 10.3390/nu13082498. PMID: 34444657; PMCID: PMC8399682.

Page 11, line 485: please rephrase the subtitle to reflect provided content

This subtitle was revised.

Page 11, lines 514-515: this sentence should be rephrased

This was re-worded.

Page 13, lines 580-582: this sentence is understandable and should be rephrased. Consider using the term eicosanoid precursor fatty acids. Additionally, throughout the whole manuscript, uniform fatty acid nomenclature (omega or n)

This section was rephrased.  The term “eicosanoid precursor fatty acid” was used.  The manuscript was reviewed, and the nomenclature “omega” was only used in tables and titles. 

Page 13, lines 594-595: add reference(s)

An existing reference was moved to this sentence.

Are there data on the doses for PUFAs used in clinical studies?

The ideal dosing of PUFAs has not been established in clinical study.  A reference was added to further describe the safety window in regards to dose. 

Agostoni C, Bresson JL, Fairweather Tait S, Flynn A, Golly I, Korhonen H, Lagiou P, Løvik M, Marchelli R, Martin A, Moseley B. Scientific opinion on the tolerable upper intake level of eicosapentaenoic acid (EPA), docosahexaenoic acid (DHA) and docosapentaenoic acid (DPA): EFSA panel on dietetic products, nutrition and allergies (NDA). EFSA JOURNAL. 2012;10(7):1-48.

Lines 621-649: replace “immunonutrition” with “immunonutrients.”

When appropriate, “immunonutrition” was replaced by “immunonutrients.”

Reviewer 3 Report

Poblete et al. presented how malnutrition impacts the biomolecular mechanisms of secondary brain injury in acute neurological disorders, and how nutritional status can be optimized in this population. This is an interesting review paper that highlights emerging nutritional therapeutics to support neurologic recovery and advanced diagnostics for nutritional assessment with investigations in brain injury. The paper is well written. There are a few issues to be addressed to further improve the manuscript.

1.     As to immunonutrition, the authors mentioned glutamine, arginine, nucleotides and their metabolites in the text. Other immunonutritions such as probiotics and prebiotics that maintain and enhance healthy intestinal immunity should also be discussed. Furthermore, other amino acids such as superoxide dismutase, glutathione peroxidase described just in Table 1, also should be discussed in the text.

2.     The authors mentioned that of greatest scientific priority, patients factors, including age, sex, race, should be delineated. In addition to these factors, nutrition during early life may influence from an acute brain injury, which should be recommended to discuss in the text.

Author Response

Poblete et al. presented how malnutrition impacts the biomolecular mechanisms of secondary brain injury in acute neurological disorders, and how nutritional status can be optimized in this population. This is an interesting review paper that highlights emerging nutritional therapeutics to support neurologic recovery and advanced diagnostics for nutritional assessment with investigations in brain injury. The paper is well written. There are a few issues to be addressed to further improve the manuscript.

  1. As to immunonutrition, the authors mentioned glutamine, arginine, nucleotides and their metabolites in the text. Other immunonutritions such as probiotics and prebiotics that maintain and enhance healthy intestinal immunity should also be discussed. Furthermore, other amino acids such as superoxide dismutase, glutathione peroxidase described just in Table 1, also should be discussed in the text.

Probiotics and prebiotics was added to Table 1 and a brief paragraph was written under the “Nutritional Therapies” section.  Specific amino acids such as superoxide dismutase and glutathione peroxidase are not commonly used in clinical practice and was therefore considered out of the scope of this article.

  1. The authors mentioned that of greatest scientific priority, patients factors, including age, sex, race, should be delineated. In addition to these factors, nutrition during early life may influence from an acute brain injury, which should be recommended to discuss in the text.

The impact and importance of nutrition during neurodevelopment and early life is discussed in an earlier section.  The paragraph in question was slightly revised to encompass “genetic and environmental factors.”